# *BARD1* and Breast Cancer: The Possibility of Creating Screening Tests and New Preventive and Therapeutic Pathways for Predisposed Women

**DOI:** 10.3390/genes11111251

**Published:** 2020-10-24

**Authors:** Marcin Śniadecki, Michał Brzeziński, Katarzyna Darecka, Dagmara Klasa-Mazurkiewicz, Patryk Poniewierza, Marta Krzeszowiec, Natalia Kmieć, Dariusz Wydra

**Affiliations:** 1Department of Gynecology, Gynecologic Endocrinology and Gynecologic Oncology, Medical University of Gdańsk, Prof. Marian Smoluchowski Str. No. 17, 80-214 Gdańsk, Poland; m.brzezinski@gumed.edu.pl (M.B.); dklasa@gumed.edu.pl (D.K.-M.); mkrzeszowiec@uck.gda.pl (M.K.); dwydra@gumed.edu.pl (D.W.); 2St. Adalbert’s Hospital, Department of Gynecology and Obstetrics, St. Jean Paul 2nd No. 50 Avenue, 80-462 Gdańsk, Poland; kasia.darecka@gmail.com; 3Warsaw College of Engineering and Health, The Battle of Warsaw 1920. Str. No. 18, 02-366 Warsaw, Poland; patryk.poniewierza@medicover.pl; 4Department of Oncology and Radiotherapy, University Clinical Center in Gdańsk, Prof. Marian Smoluchowski Str. No. 17, 80-214 Gdańsk, Poland; natalistro@gmail.com

**Keywords:** breast cancer, BARD1, surveillance, management, genetic testing, predisposition, susceptibility, neoadjuvant, chemotherapy

## Abstract

Current oncological developments are based on improved understanding of genetics, and especially the discovery of genes whose alterations affect cell functions with consequences for the whole body. Our work is focused on the one of these genes, *BRCA1*-associated RING domain protein 1 (*BARD1*), and its oncogenic role in breast cancer. Most importantly, the study points to new avenues in the treatment and prevention of the most frequent female cancer based on *BARD1* research. The *BARD1* and *BRCA1* (BReast CAncer type 1) proteins have similar structures and functions, and they combine to form the new molecule *BARD1*-*BRCA1* heterodimer. The *BARD1*-*BRCA1* complex is involved in genetic stabilization at the cellular level. It allows to mark abnormal DNA fragments by attaching ubiquitin to them. In addition, it blocks (by ubiquitination of RNA polymerase II) the transcription of damaged DNA. Ubiquitination, as well as stabilizing chromatin, or regulating the number of centrosomes, confirms the protective cooperation of *BARD1* and *BRCA1* in the stabilization of the genome. The overexpression of the oncogenic isoforms *BARD1β* and *BARD1δ* permit cancer development. The introduction of routine tests, for instance, to identify the presence of the *BARD1β* isoform, would make it possible to detect patients at high risk of developing cancer. On the other hand, introducing *BARD1δ* isoform blocking therapy, which would reduce estrogen sensitivity, may be a new line of cancer therapy with potential to modulate responses to existing treatments. It is possible that the *BARD 1* gene offers new hope for improving breast cancer therapy.

## 1. Introduction

In 1996, Wu et al. discovered a binding partner protein of *BRCA1* (BReast CAncer type 1) which they named *BRCA1*-associated RING domain 1 (*BARD1*) [1]. The *BARD1* gene is located on chromosome 2 and encoded by the sequence 2q34-q35. Its product is a 777 amino acid protein composed of an N-terminal RING-finger domain, three Ankyrin repeats (ANK) domains, and two tandem *BRCA1* C-terminal (BRCT) domains (Figure 1). The *BARD1* protein structure is like that of the *BRCA1*, however it is different from that of the *BRCA2* (BReast CAncer type 2), the second gene associated with breast cancer [2]. *BARD1* and *BRCA1* can form a heterodimer by their N-terminal RING finger domains which form a stable complex [3]. The full length-*BARD1* (FL-*BARD1*) protein has tumor-suppressor functions whether it acts as a heterodimer or in *BRCA1* independent pathways. However, the aberrant splice variants of *BARD1* have oncogenic functions. The two major isoforms involved in the breast cancer pathogenesis are *BARD1β* and *BARD1δ* [4].

The *BARD1*-*BRCA1* heterodimer, as an E3 ubiquitin ligase, is essential in numerous cell regulations [4]. Its primary function is to allow ubiquitin to be attached to different proteins which marks them for further degradation. Due to this ability, the *BARD1*-*BRCA1* heterodimer is engaged in the DNA damage response pathway [4]. Its BRCT motifs are phosphoprotein-binding modules and bind to poly(ADP-ribose) (PAR), which targets the *BARD1*-*BRCA1* heterodimer to DNA damage sites, where it acts as an E3 ubiquitin ligase. As a result, *BRCA1* is able to participate in all major DNA repair pathways [5,6].

Moreover, this heterodimer prevents the transcription of the damaged DNA and maintains its genetic stability by ubiquitinating RNA polymerase II [7].

The *BARD1*-*BRCA1* heterodimer is also responsible for the ubiquitination and subsequent degradation of estrogen receptors α (ERα). It is an important function in terms of pathogenesis of breast cancer as estrogen receptors α (ERα) and β (ERβ) activate genes responsible for cell proliferation [8].

*BARD1* is also able to function as a protein in the *BRCA1*-independent pathways. *BARD1* has a crucial role during the induction of apoptosis by the stabilization of p53 [9]. Likewise, it inhibits mRNA maturation during genotoxic stress through having an impact on CstF-50 (cleavage stimulation factor) [10].

All these functions prove that FL-*BARD1* has an important tumor suppressor role. However, in neoplastic pathogenesis, *BARD1* isoforms antagonize FL-*BARD1* and enable uncontrolled proliferation. The main cancerous isoforms are *BARD1β* and *BARD1δ*.

The first of the isoforms, *BARD1β*, stabilizes Aurora kinases A and B. It forms a complex with *BRCA2* and Aurora B during telophase and cytokinesis that results in overriding the mitotic checkpoint and excessive cell proliferation. Thus, Aurora family of kinases and *BARD1β* expression levels might be predictive biomarkers for responses to Aurora inhibitors [11]. The second key isoform, *BARD1δ*, interacts with ERα and antagonizes FL-*BARD1* that results in a higher response rate to estrogens [8].

Breast cancer is the second most common neoplasia in the female population. Despite this fact, no more than 40% of familial breast cancers have been identified as having causative gene mutations [12]. Most of these mutations are in either the *BRCA1* or the *BRCA2* genes. The latest reports show though that deleterious *BARD1* variants may be the reason for hereditary breast cancer in *BRCA1* and *BRCA2* negative families [13]. There are already available new types of tests that show the presence of mutations not only in the *BRCA1* or *BRCA2* gene, but also in *BARD1*. *BARD1* seems to be an interesting target for novel therapies as it is involved in many different cellular processes and therefore it has a lot of potential therapeutic targets.

The *BARD1* protein also seems to be an interesting starting point in analyzing the causes of drug resistance in breast cancer cases. About 70% of breast cancers are ER positive. Despite using multiple drugs that are ER antagonists (e.g., tamoxifen) we still observe numerous relapses, even during 15 years of post-treatment follow-up [14]. The main limitation in solving this problem is that the mechanisms of chemoresistance are still too-little understood. However, it seems that the *BARD1* protein, that is associated with so many cellular mechanisms, can play a key role here [14].

The aim of our review is to investigate the role of the *BARD1* gene in the assessment of predisposition to breast cancer, which is related to the question of the usefulness of testing this gene in screening programs for families with familial history of breast cancer, and further, to investigate it as a potential target of new anticancer therapies, including sensitivity to chemotherapy.

## 2. Scope of the Review

The article reviews the literature using the Pubmed, Google Scholar and Elsevier Clinical Key databases using the terms: “breast cancer”; “BARD1”; “surveillance”; “management”; “genetic testing”; “predisposition”; “susceptibility”; “neoadjuvant”; “chemotherapy” in various combinations as appropriate. Articles were screened for relevance, those with the most up-to-date information were selected for inclusion. In addition, a manual search of the reference lists of previously captured articles was carried out to increase the likelihood of choosing essential studies.

## 3. The Results of the Review on the Topics Covered

We focused on four main issues regarding *BARD1*. The first one discusses the frequency of mutations in the *BARD1* gene in non-*BRCA1* and non-*BRCA2* patients with breast cancer. It shows that *BARD1* is one of the most common non-*BRCA1/2* genes to mutate. For this reason, subsequently we present different possibilities for running a surveillance program for *BARD1* for example detection of *BARD1* gene isoforms by using specific antibodies or radiogenomics, which link clinical assessment with imaging results and genetic background. The next part discusses the possibility of using *BARD1* as a target for new therapies using drugs such as an inhibitor of CDKs, mTOR inhibitor, PI3K inhibitors or PARP inhibitors (inhibitors of the enzyme poly ADP ribose polymerase) and one of the histone deacetylase inhibitors. Finally, we consider *BARD1* gene mutations and neoadjuvant setting in breast cancer which is an important medical treatment modality for breast cancer patients treated today.

### 3.1. The Significance of *BARD1* in Genetic Predisposition to Breast Cancer

Genetic predisposition to breast cancer can be divided into three different levels [15,16], depending on the risk of breast cancer and the degree of gene penetrance. The first level is composed of high-risk heterozygous, and highly penetrant gene mutations. The second level is associated with genes of intermediate penetrance and a moderate risk of breast cancer. The third level consists of low-penetrance breast cancer susceptibility alleles, and common polymorphisms (SNPs—single nucleotide polymorphisms) [13,16] (Table 1).

Since its discovery in 1996, the *BARD1* gene and its various mutations have been extensively studied for breast cancer susceptibility. In a study of over 65,000 American non-*BRCA1* and non-*BRCA2* patients (mean age at diagnosis 48.5) with breast cancer, pathogenic variants in *BARD1* in white women were associated with a significant moderately increased risk of breast cancer. The pathogenic variant (PV) in this population, proved to be quite rare (<1 out of 500 breast cancer cases) [20].

*BARD1* is not only thought to be a breast cancer susceptibility gene, but also a gene predisposing to triple negative breast cancer (TNBC) [21]. Furthermore, in a study of 10,901 TNBC patients, it was established that *BARD1* was one of the most common non-*BRCA1/2* genes to mutate. Among other genes [21], *BARD1* was proven to be statistically significantly associated with a moderate to high risk of TNBC with an incidence of 0.5–0.7% [21]. The same study established that the PVs in *BARD1* were associated with a lifetime risk of TNBC in 7% of cases; and a 21% risk for Caucasian patients and 39% risk of TNBC for African American patients [21]. In a different study of 289 African American patients, 144 of whom were cases of familial breast cancer, only one incidence of PV in the *BARD1* gene was found [22]. In another study of 1824 female American patients with TNBC, 97% of which were white, 1.9% African, 0.6% Asian, and 0.6% Hispanic, deleterious mutations in *BARD1* were detected nine times, with an incidence of 0.3–0.5% [23].

Outside the United States, there has also been research on *BARD1* in Europe, Korea, and Australia. Out of 120 Korean breast cancer patients negative for *BRCA1/2* mutations, PVs in the *BARD1* gene were identified in two patients [24]. A Finnish study of 94 *BRCA1/2* negative breast cancer families, established an incidence of 7.4% of Cys557Ser allele in the *BARD1* gene in comparison with an incidence of 1.4% in the healthy controls [25]. Moreover, the *BARD1* Cys557Ser allele was also reported in an Italian study with an incidence of 2.5% [26]. These studies may indicate that the *BARD1* Cys557Ser allele is of European origin.

In three independent studies of the Polish population, a deleterious nonsense pathogenic *BARD1* mutation, namely p.Q564X, was identified [13,27,28]. A study among 12,476 Polish and 1459 Belarusian breast cancer patients, identified a 0.27% incidence of the PV in both study groups, assessing it as a low/moderate breast cancer predisposition gene. The p.Q564X *BARD1* mutation is possibly a founder mutation, present at least in Central Europe. However, its presence in the Polish control subgroup (0.15%) might indicate its low penetrance. It is also important to point out, that a higher incidence of the mutation was found in progesterone receptor-negative breast cancer patients than in the group of receptor-positive breast cancer patients (0.55% and 0.24%, respectively) [29]. An analysis of large mutations of the *BARD1* gene in 504 breast cancer/ovarian cancer Polish patients was conducted and indicated that such mutations do not contribute to breast cancer predisposition [13]. This, however, does not contradict the role of *BARD1* as a breast cancer susceptibility gene.

A study in Germany, inspecting germline loss-of-function (LoF) variants in the *BARD1* gene, was conducted in 4469 breast cancer patients, 23 (0.51%) of whom had LoF variants. Those patients were significantly younger at first diagnosis than in the overall population sample (median age 42.3 vs. 48.6, respectively). LoF *BARD1* variants were not significantly associated with patients with age at first diagnosis of equal or older than 50 years. This might suggest a need to intensify breast cancer surveillance programs and include testing for *BARD1* PV [30].

However, controversy remains as to whether the *BARD1* variant, in its rarity, can be clinically associated with increased breast cancer risk [24]. There have also been studies disputing that *BARD1* is a moderate/high-risk breast cancer susceptibility gene [31]. In a study of 684 Australian non-*BRCA1/2* patients with familial breast cancer, four cases of PVs in *BARD1* were identified (0.6%), and the study concluded there is no clinical value for the *BARD1* PV mutation testing in breast cancer families.

### 3.2. Utility of *BARD1* in Surveillance Programs

Currently, *BRCA1*/*BRCA2* is the best-known gene relating to breast cancer. Depending on their age, those carriers at high, or very high-risk need: regular breast self-examination, imaging such as mammography or breast magnetic resonance imaging (MRI) every 6–12 months, transvaginal ultrasound every 6 months, and CA-125 blood testing due to the increased risk of ovarian cancer [32,33]. Patient monitoring can also include prophylactic mastectomy and bilateral salpingo-oophorectomy; though these procedures severely affect the patient’s quality of life and can hamper her psychosocial well-being as a result of infertility [34,35]. Bearing this in mind, it seems justified that stricter monitoring should be undertaken, including risk stratification based on genetic testing. The *BARD1* gene appears beneficial for patient observation. Based on 2019 study of a group of 4469 women, it was concluded that the *BARD1* gene correlates with early onset of breast cancer and a worse prognosis [30,36]. The mutated gene carriers should be screened at a younger age, especially because the gene has also been shown to be related to other cancers, including ovarian cancer, colorectal cancer, non-small-cell lung carcinoma, and hepatocellular carcinoma [37,38]. The breast cancer cells produce isoforms of the *BARD1* gene, which can be detected with specific antibodies [39,40,41]. Interestingly, the isoforms can also be produced by spontaneous breast cancer not associated with *BRCA* group genes [42]. The above phenomenon may correlate with the fact that *BARD1* functions as a factor of apoptosis, unrelated to *BRCA1*. Isoforms excessively expressed in tumor cells do not have suppressor functions, which leads to faster progression and poor prognosis [4,43,44].

A study published in 2007 found *BARD1* isoforms to have a distinctive expression pattern. The full-length isoform accounted for 0%, while splice isoforms associated with alternative transcription initiation in exon 4—for as much as 80.8% in different breast cancer cell lines (21 out of 26) [39].

In addition to the most obvious role of a screening test, the antibody testing can also be used for treatment monitoring because the increased expression of *BARD1* isoforms is associated with disease progression. Immunohistochemical testing of breast cancer samples shows more intense staining of the cytoplasm due to the overexpression of *BARD1* isoforms. It is worth mentioning that the degree of staining was proportional to the degree of malignancy and size of the tumor. Comparing those observations with the Tumor-Nodes-Metastasis staging system, it is hypothesized that overexpression of *BARD1* isoforms is proportional to the size of the tumor and its malignancy grade, which in turn heralds a worse prognosis [44,45,46]. Recently, there has been a growing body of research suggesting that the *BARD1* gene is only associated with low to intermediate risks of breast cancer [29]. Other genes, such as *PALB2*, *BRIP1*, *ATM*, *CHEK2*, *RAD51C*, *RAD51D*, *NBN*, *NF1*, and *MMR* should also be considered, as these can have a cumulative effect on the risk of breast cancer in combination with the *BARD1* gene [19,46]. Our observations made here strongly advocate for patient surveillance based on multigenetic panel testing, or even for an individualized approach and monitoring based on genetic profiling [47].

Mammography and breast MRI remain the fundamental imaging modalities for the high and very high-risk patients. Decreasing mortality rates are thought to have resulted from more effective treatment [48]. Therefore, other diagnostic tools should be sought, not only for screening but also for risk management. An interesting approach might be radiogenomics, which brings together clinical assessment, imaging results, and genetic background [49]. This approach would be of interest in relation to the immunohistochemical staining of the *BARD1* gene, which in turn can be imaged in magnetic resonance scans. The precise diagnosis may play a role in decisions about whether to perform or postpone prophylactic surgical interventions due to breast cancer risk. However, the multidirectional diagnostic pathway as a standard approach requires further cohort trials and can be of interest for future researchers.

### 3.3. *BARD1* Gene as a Potential Target of New Anticancer Therapies Including Sensitivity to Chemotherapy with a Focus on Breast Cancer

Several studies have shown that *BARD1* can potentially become a new target for breast cancer treatment. Zhu Y et al. [14] have reported that the significantly higher expression of *BARD1* and *BRCA1* in tamoxifen-resistant breast cancer cells results in resistance to DNA-damaging chemotherapy with cisplatin and adriamycin, but not with paclitaxel. While the mutations of *BRCA1* and *BARD1* cause a defective DNA damage repair, they also lead to increased sensitivity to platinum-based chemotherapy. The authors have also suggested that the consideration of microtubule-targeting agents such as taxanes while planning chemotherapy for tamoxifen-resistant breast cancer patients may be superior to DNA-damaging agents (i.e., anthracyclines and platinum compounds). Additionally, they have demonstrated that silencing the gene expression of the aforementioned proteins using siRNAs or phosphorylation inhibition of *BRCA1* by a CDK inhibitor, dinaciclib restores the sensitivity to cisplatin in those cells. Since the simultaneous silencing of *BARD1* and *BRCA1* have failed to show any addictive effect, they have deduced that effects of the therapeutic inhibition are propagated via the same pathway [14]. The same authors have also shown that PI3K inhibitors decrease the expression of *BARD1* and *BRCA1* in tamoxifen-resistant cells and resensitize them to cisplatin, both in vitro and in vivo. Hence, they have concluded that the PI3K/Akt/mTOR pathway is responsible for the upregulation of *BARD1* and *BRCA1*. This intracellular signaling pathway is responsible for the control of proliferation, apoptosis, angiogenesis, and cell survival. The mutations affecting this pathway are the most encountered genetic alteration in ER-positive breast cancer, as well as in recurrent or metastatic cancers [50]. They also increase the activation of the PI3K/Akt/mTOR pathway, which influences the resistance to hormonal cancer therapy [51].

Li M et al. [5] have reported that the *BARD1* BRCT domain interacts with poly(ADP-ribose) (PAR), which results in subsequent recruitment of the *BARD1*-*BRCA1* complex to the damaged DNA.

The poly ADP-ribosylation (PARylation) is of particular importance since the promising drugs inhibiting the PAR polymerizing enzyme (PARP) appear to be more efficient in *BRCA1*-mutated cells with preserved *BARD1* tumor suppressor function. The PARylation serves as a signal to recruit DNA damage repair proteins such as the *BARD1*-*BRCA1* complex to the double-strand breaks (DSBs). The *BARD1* BRCT domain by binding ADP-ribose, a basic unit of PAR, recruits *BRCA1* to the DNA damage sites. This recruitment resulting in formation of *BARD1*-*BRCA1* heterodimer can be suppressed by the PARP inhibition, which selectively eliminates *BRCA1*-deficient cells. Several PARP inhibitors (PARPi) have recently been approved by the FDA for the treatment of various neoplasms, including metastatic TNBC and estrogen receptor-negative (ER-)/HER2+ breast cancer with *BRCA* mutations [52].

Throughout treatment both ovarian and breast cancer patients harboring *BRCA1* mutations initially responding to the platinum and PARPi therapy, develop the resistance to both PARPi and platinum compounds [53,54,55]. This resistance, as examined by the patient’s biopsies could result from the observed secondary mutations or the methylation status of *BRCA1*, *BRCA2*, and other genes controlling the homologous recombination. One potential way to overcome this resistance could be the investigation of whether the expression of FL or the isoform of *BARD1* contributes to the success or failure of the PARPi therapy [53].

None of above-mentioned inhibitors specifically affect *BARD1*. Lepore et al. [56] have shown that Vorinostat, a histone deacetylase inhibitor (HDACi) lowers the *BARD1* isoform mRNA levels through increased miR-19a and miR-19b expression. Additionally, they have reported that the expression of the truncated *BARD1* isoforms expressed in human acute myeloid leukemia (AML) cell lines is modulated by HDACi treatment via miR-19a/b. To verify whether this was an exclusive event to human AML cell lines, they have evaluated Vorinostat-induced downregulation of *BARD1* expression in additional human cancer cell lines, such as MCF7 breast cancer cells, HeLa cervical cancer cells, and Kelly neuroblastoma cells. The time-dependent reduction has been observed in Kelly and MCF7 cell lines, but not in HeLa lines, indicating that the *BARD1* dysregulation is cell line-restricted. Interestingly, cells affected by Vorinostat weakly express FL *BARD1* [56].

### 3.4. *BARD1* Gene Alterations in Neoadjuvant Setting in Breast Cancer

Current data on this subject mainly refer to TNBC which has higher incidence of pathogenic variants of the *BARD1* gene [57]. Watanabe et al. analyzed 30 TNBC core biopsy specimens of patients with pathologic complete response (noninvasive cancer) and noncomplete response following neoadjuvant chemotherapy (NACT), with regard to the aberrant DNA methylation status of the *BARD1* gene (from a total number of 16 DNA repair genes) using bisulfite-pyrosequencing. Although hypermethylation of the *BRCA1* gene is associated with TNBC subtype and may impact chemosensitivity and progression under NACT, *BARD1* gene hypermethylation revealed only a low-to-moderate influence on these processes [58]. Some other studies underline the low incidence and uncertain clinical impact of gene mutations other than *BRCA1/2* (including *BARD1*), and the associated unfavorable outcomes for patients with breast cancer undergoing NACT [59]. Yet other studies reported that *BRCA1* and its associated protein *BARD1* are upregulated in tamoxifen-resistant breast cancer cells, rendering the cells resistant to DNA-damaging chemotherapy [14,60]. Today, neoadjuvant chemotherapy makes a significant contribution to chemotherapy in breast cancer and is a bridge towards adjuvant regimens and other therapies. Intensifying research into the role of *BARD1* in chemotherapy in women for whom NACT is planned is essential and should be of benefit.

## 4. Discussion

The U.S. National Comprehensive Cancer Network (NCCN) guidelines do not routinely recommend *BARD1* positive patients to undergo additional breast cancer screening (early breast MRI, mammography), which might need to be implemented [21]. This screening is usually only performed in the cases with family history indicating that the patient has an increased risk of breast cancer. For now, the risks of breast cancer connected with *BARD1* remain poorly defined and of varying prevalence across different populations. Nonetheless, there is multiple instances, listed in the evidence above, that the PVs of *BARD1* not only increase the risk of breast cancer in general, but primarily of TNBC, and can be associated with age at first diagnosis of equal or under 50 years.

The *BARD1* gene can significantly extend the monitoring options in patients at risk of breast cancer and during post-treatment follow-up. However, due to the low incidence of *BARD1* mutations, such assumptions require further long-term population-based trials. At present, and considering potential benefits and costs, it seems that the possible uses of the *BARD1* gene that we discussed in this review can set a direction for further research rather than provide real options for widespread use. However, it should be noted that commercial tests are available that can detect a mutation in the *BARD1* gene.

Over the last few decades, molecular research has been intensified to further individualize the treatment of breast cancer patients. Personalization of systemic treatment is aimed at identifying a group of patients with unfavorable prognostic factors and at identifying patients who can benefit most from therapy [47]. The assessment of efficacy of PARPi in breast cancer patients with the relatively frequent LoF mutations of *BARD1* would be of necessity to improve patients’ outcomes. Ozden et al. [61] proved that *BARD1β* sensitizes colon cancer cells to poly PARP-1 inhibition even in an FL *BARD1* background, thus suggesting that *BARD1β* may serve as a future biomarker for assessing the suitability of colon cancers for homologous recombination targeting with PARPi in the treatment of advanced colon cancer. Clinical trials of PARPi in neoadjuvant, mono-, and combination therapy settings in breast cancer are ongoing.

Neoadjuvant chemotherapy offers opportunity to assess the molecular changes of heterogenic breast cancer tissue before and after chemotherapy, especially in the case of TNBC, in which *BARD1* gene deleterious alterations are the most prevalent and NACT seem to have the greatest value.

Near half of cited papers about breast cancer in this review did not relate to any molecular subtype. Most of the relevant studies suggest the role of *BARD1* in breast cancer is in TNBC patients. This shows that more research is based on, and needed into, the genesis and therapeutic potential of TNBC.

## 5. Conclusions

Analyzing structure and functions of the *BARD1* gene, we believe that *BARD1* gene can play an important role in the pathogenesis of breast cancer and in the mechanisms of chemo-resistance of cancer cells as well.

It is reasonable to screen *BARD1* gene isoforms in certain populations, especially in those with evidence of higher prevalence of mutations in the *BARD1* gene. This approach would also have to be researched for its relevance to general breast cancer patient outcomes, survival rates, quality of life, influence on treatment decisions, and cost-effectiveness.

Radiogenomics is a promising field of science as a bridge between molecular and imaging medicine. Broader prospective studies and standardization (i.e., immunohistochemistry studies with *BARD1*-directed antibodies) will provide determination of appropriate imaging biomarkers enabling “cancer cell visibility” before they can be introduced into a clinical investigation.

Further research on the *BARD1* gene expression may contribute to the effective reversal of PARPi resistance and the wider introduction of new targeted therapies for the treatment of breast cancer patients.

Data on patients with *BARD1* gene polymorphism undergoing NACT for breast cancer are limited. However, gene expression alterations after NACT can shed light on the pathogenesis of this multifactorial disease.

## Figures and Tables

**Figure 1 genes-11-01251-f001:**
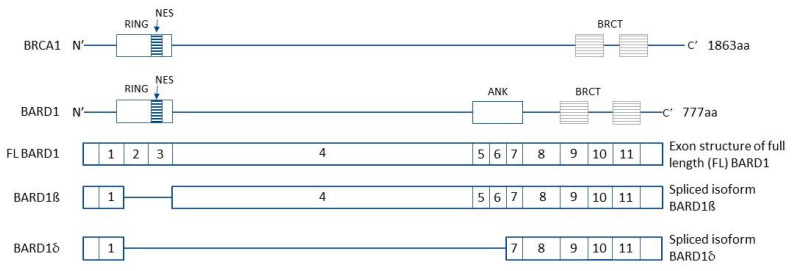
Schematic structures of *BRCA1*, *BARD1*, and isoforms: *BARD1β*, *BARD1δ*. RING finger domains enable to form a stable complex between *BRCA1* and *BARD1*. NES are nuclear export signals which together with NLS (nuclear localization signals) are necessary for proper intracellular localization of *BARD1*. ANK (Ankyrin repeats) interacts with several proteins including p53 and NF-κB. BRCT (*BRCA1* carboxy-terminal domain) motifs fold into a binding pocket with a key lysine residue (K619).

**Table 1 genes-11-01251-t001:** Levels and characteristic of genetic predisposition to breast cancer.

Level of Predisposition	Gene Penetration	Risk of Breast Cancer	Examples of Affected Genes	Characteristics	Reference
**I**	High	High	*BRCA1* and *BRCA2*, *TP53*, *CDH1*, *STK11*, *PTEN*	Mutations in *BRCA1* and *BRCA2* are responsible for 16–40% of hereditary breast and ovarian cancers and site-specific breast cancer; in*TP53* is associated with up to 85% risk of developing breast cancer by age 60; germline mutations in *CDH1* and *STK11* are associated with 39–52% and 32–54% risk of developing breast cancer, respectively; germline mutations in the *PTEN* gene promoter are associated with an 85% lifetime risk of breast cancer	[12,17,18,19]
**II**	Intermediate	Moderate	*ATM*, *CHEK2*, *BRIP1*, *BARD1*, *PALB2*	Mutations in these genes are responsible for a 2- to 4-fold increase in the risk of breast cancer in comparison to population-based risk	[16]
**III**	Low	Low	*FGFR2*, *RAD51*	*FGFR2* SNPs increase the risk of breast cancer by increasing the response to estrogen; *RAD51* *SNP2* i.e., are considered as *BRCA1/2* mutation carrier risk modifiers	[13,16]

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
