# Peer review of "BARD1 and Breast Cancer: The Possibility of Creating Screening Tests and New Preventive and Therapeutic Pathways for Predisposed Women"

_genes, 2020, doi:10.3390/genes11111251_

Round 1
Reviewer 1 Report
All my suggestions have been addressed.
Reviewer 2 Report
corrections made from last version are ok
This manuscript is a resubmission of an earlier submission. The following is a list of the peer review reports and author responses from that submission.
Round 1
Reviewer 1 Report
This manuscript by Sniadecki et al, is a review of BARD1 in heritable breast and ovarian cancer, and this manuscript discusses screening tests and the potential for therapy. The ideas in this review are interesting, and not covered to my knowledge in other reviews (or many studies), but these ideas need to be developed much better for this to be an excellent review paper. More specific information needs to be provided and concepts better developed. Plus, a key function of review articles is to provide important references for the subject to a reader that is potentially new to the field, but there are many paragraphs in this review paper that are wholly without citation.
The focus of this article is BARD1 and the gene as a target for genetic screening and potential therapies. BARD1 has few pathogenic variants that cause disease, and the authors develop in the abstract and in the main body the concept that overexpression of splice variants of BARD1 that lack the RING domain and function as dominant-negative versions of the protein can be important in oncogenesis. This is a very interesting idea, but it needs to be better supported. First, the only citation for this concept is another review article, and that is not adequate or appropriate. Second, this review needs to be specific about how many tumors this overexpression is detected in, the frequency of expression of these splice variants, the expression of these variants in other tumor types, etc. It is possible that this proposed mechanism of BARD1-isoform mediated oncogenesis explains why pathogenic variants are rarer in BARD1 than in BRCA1 or BRCA2. This must be better developed with more specific findings and more citations of original research articles.
The second concept of interest is inhibition of BARD1 as a therapeutic. All of the targets described are indirect or nonspecific. Plus, one must discuss the logic of inhibiting a tumor suppressor, which is usually a bad idea, but maybe it should be considered if splice variants are overexpressed.
There are a number of specific issues listed below in addition to the broad concerns discussed above:
- Line 20: “the most important of these genes, the BARD1..” It is hard to argue that BARD1 is the most important gene. It is an important gene, certainly not the most important gene.
- Line 25: I am unaware how BARD1-BRCA1 destroys damaged DNA fragments. It regulates a pathway that chews back a strand of DNA from a site of double stranded break but it does not catalyze the DNA destruction.
- Line 29: it would be useful to cite here how frequent the beta and delta isoforms of BARD1 are detected as overexpressed.
- Line 53 and line 58; what is meant by “motifs combine with a key lysine residue (K619)..” Isn’t K619 part of the BRCT motif?
- Lines 60-61: Is it known that BARD1-BRCA1 ubiquitination of RNA polymerase II is part of the DNA damage pathway? I do not think that original research showed that.
- Lines 56-61: these concepts need to be well-cited.
- Lines 68-71: need citations!
- Lines 79-85: need multiple citations!
- Lines 178-181: This section introduces the concept of splice isoforms of BARD1 acting as dominant-negative versions of BARD1 and contributing to oncogenesis. This is an important concept since it is in the abstract, and interestingly might explain why pathogenic variants of BARD1 are much more rare than BRCA1 pathogenic variants. This point needs to be supported with citations of the original literature, not just a review article, and it is important that the incidence/frequency of these overexpressed splice isoforms be discussed (with original citations).
- Line 237-238: “PARP inhibition suppresses the recruitment of the BARD1-BRCA1 heterodimer to DNA damage sites and impairs DNA repair.” This sentence is potentially misleading since it implies a connection between the BARD1-BRCA1 recruitment to damage sites and inhibition of DNA repair, but PARP inhibitors affect DNA repair in multiple ways.
- Section 3.3, lines 204-248: None of these inhibitors specifically affect BARD1. Are there any that would uniquely target a BARD1 splice isoform?
Reviewer 2 Report
The review is not well written as there is not logic structure from the beginning of the manuscript and the authors are just listing the conclusions from various papers without his own understanding and orgnization. Also, a good previous review with title" New concepts on BARD1: Regulator of BRCA pathways and beyond" have covered most of points in current manuscript. Furthermore, the authors do not read the papers careful enough: For example, the length of BRCA1 is wrong, it should not be 1823 aa; The number of ANK repeats in BARD1 is wrong; BARD1 has six NLS? There are papers have further demonstrated only a few NLS is real. As such, the review is not at all for publication.
Reviewer 3 Report
The review titled “BARD1 and breast cancer: the possibility of creating screening tests and new preventive and therapeutic pathways for predisposed women” by Sniadecki et al is interesting and is within the scope of “Genes” Journal. But, it needs revision.
- Abstract: “Our work is focused on the most important of these genes, the BARD1 and its oncogenic role in breast cancer.”
The line is comes out to be over exaggeration. Please reword the sentence.
2) The introduction of routine tests, for instance, to identify the presence of the BARD1β
…..gene offers new hope for improving breast cancer therapy.
The font is different than rest of the font of the article.
3) In general a review article does not need material and methods or results section. Logical headings are needed to define the flow of the article.
4) Most of the relevant studies suggesting the role of BARD1 in breast cancer is in Triple negative breast cancer patients. Suggestion that instead of stating BARD1 in breast cancer, the authors should consider writing the title as BARD1 in triple negative breast cancer.
5) The review is not able to stress what added advantage of screening BARD1 will be over BRCA1? All the literature cited describes a very preliminary role of BARD1 and does not convince the readers that BARD1 will be a novel molecule for creating screening tests.
Round 2
Reviewer 1 Report
Lines 200-202: splice isoforms account for 80.8% of what? Please clarify. Is it 80.8% of tumor samples? While that is being implied, that would seem rather high. Also, exon 4 skipping would retain the RING domain, and I think loss of the RING domain is what would be critical for dominant negative activity. Please clarify and/or comment.
Please delete Table 2 (which is new in the revision)
Reviewer 2 Report
The review is not well written as there is not logic structure from the beginning of the manuscript and the authors are just listing the conclusions from various papers without his own understanding and organization. Again, the authors do not read the papers careful enough: For example, BARD1 has six NLS? There are papers have further demonstrated only a few NLS is real. As such, the review is not at all for publication.